# Machine learning for detecting age and sex effects on corpus callosum volume

Handan Soysal[1], Niyazi Acer[2], Meltem Özdemir[3] and Kazim Gumus[4]

[1] Department of Anatomy, Faculty of Dentistry, Ankara Yıldırım Beyazıt University, Ankara, Turkey
[2] Department of Anatomy, Faculty of Medicine, Sanko University, Gaziantep, Turkey
[3] Department of Radiology, Dışkapı Yıldırım Beyazıt Health Application and Research Center, Medical Sciences University, Ankara, Turkey
[4] Department of Radiology, University of Florida, Florida, United States

## ABSTRACT

**Aim:** This study aims to investigate whether machine learning (ML) could identify the volumetric changes in the corpus callosum (CC) and its sub-regions (genu, body, and splenium) due to aging and sex differences.

**Material and Methods:** Brain magnetic resonance imaging (MRI) images obtained from 301 healthy male and female subjects were used in the study. The volume measurements of the corpus callosum and its sub-regions were calculated from MRI images using MRICloud software. The classifications of age (young/adult) and sex (female/male) were performed using the Classification Learner Tool in MATLAB 2020b (MathWorks, Natick, MA, USA). Classifiers including k-Nearest Neighbor (KNN), support vector machine (SVM), decision tree, naïve Bayes, logistic regression, and Ensemble classifiers were evaluated using 10-fold cross-validation.

**Results:** Machine learning classification revealed moderate accuracy in distinguishing sex (best: Fine Gaussian SVM, 65.4% accuracy, AUC = 0.60) and higher accuracy in distinguishing age groups (young/adult; best: Fine Gaussian SVM, 83.7% accuracy, area under the curve (AUC) = 0.67) based on corpus callosum and its sub-region volumes derived from MRI.

**Conclusion:** This preliminary data suggests that ML can provide indications to assess the impact of age and sex on corpus callosum volume based on MRI data. The observed classification accuracies, particularly for sex, suggest that larger datasets are needed to enhance the accuracy of these models in future investigations.

## INTRODUCTION

The corpus callosum (CC) is the most important and biggest commissure and transfers data between the two hemispheres. It is divided into four distinct regions anatomically: the rostrum, genu, body, and splenium (*Battal et al., 2010*). CC could change in size due to aging. Studies have shown that CC development increases rapidly during infancy, while there is a gradual increase during youth. When the development of CC is examined with age, although development is complete by age 4, growth continues at a much slower rate until the thirties (*Soysal et al., 2022*). The CC is the last part of the brain to mature, and its development continues from late adolescence to early adulthood (*Pujol et al., 1993*). It is



Corresponding author
Handan Soysal,
handan_soysal@hotmail.com

known that volumetric changes in the CC can be associated with axonal structure, cellular changes, and the number of fibrils in this region (*Agdanlı et al., 2022*). Previous neurodevelopmental studies showing that brain maturation, especially of the corpus callosum, continues into the mid-twenties (*Genc et al., 2018*). It has been shown that there are significant relationships between the corpus callosum and age. Decreases in corpus callosum volume, fiber count, fiber length, and average fiber count per voxel have been shown to occur in early onset (mid-20s) and later onset (mid-50s) cases (*Pietrasik et al., 2020*).

Researchers often use magnetic resonance imaging (MRI) to study CC morphology, because of its high spatial resolution, multiplanar reconstruction, and involves no radiation. The quantitative volumetric analysis of CC is often performed using T1-weighted MRI images. Recently, several quantitative MRI studies were reported on the volume variation of CC with aging and gender differences (*Tanaka-Arakawa et al., 2015*; *Arda & Akay, 2019*). *Soysal et al. (2022)* reported that volumes of CC and its parts were significantly higher in males than females and also, genu and splenium volumes were observed significantly varied for both sexes in the left hemisphere. Contrarily, *Agdanlı et al. (2022)* reported no statistically significant associations between CC volume and age/gender. It was shown that there is a negative relationship between age and the thicknesses of the genu and all the body parts of CC (*Arda & Akay, 2019*). However, it was also shown in a prior study that callosal width measurements did not correlate with either age or gender (*Sullivan, Rosenbloom & Desmond, 2001*), did not show any gender difference (*Karakaş et al., 2011*), and also did not yield a significant gender difference in CC height and length, total callosal area, and thickness of different CC portions (*Ng et al., 2005*). Splenium atrophy was observed in the early stage of Alzheimer's disease (AD) (*Hampel et al., 2002*), while a decrease in genu size was observed in the normal aging process. CC size has been reported to decrease with age in men (*Witelson, 1989*). Despite some conflicting results in the literature, overall findings indicate changes in the volume of CC based on aging and sex differences which necessitates further confirmative studies.

We hypothesized that the interactions between sex and age would be specific for distinct corpus callosum, with steeper decreases with age. We used corpus callosum volumes to examine age sex interactions as well as sex influences on compartmental volumes using machine learning (ML) approach and MRICloud, an unbiased automated segmentation tool. These changes should represent sex-dependent age effects in the context of life span changes, as well as corpus callosum volume differences and sex-specific vulnerabilities in the process of aging. The number of studies investigating gender and age-related differences in the volume of the corpus callosum and its parts are insufficient. There is also no ML study in literature that examines the relationship between corpus callosum and its part volumes and age and gender. The objective of the present study is to assess the ML algorithms in learning the relationship between the volume of CC and age and sex.

## MATERIALS AND METHODS

### Participants

The study group consisted of healthy volunteers with no history of neurological or psychiatric disease, surgical treatment or trauma of the brain, or substance abuse. Participants were given a Mini-Mental State Examination (MMSE) of questionnaire. Those with impaired mental scores were not included in the study. A total of 301 cases including 117 males and 184 females, with ages ranging from 11–84 years were enrolled in the study.

### MRI data acquisition

The imaging of the participants was performed on a 1.5 T unit (Magnetom Aera, Siemens, Erlangen, Germany) with a 20-channel head coil. The MRI protocol included 3D T1-weighted, T2-weighted, and diffusion-weighted imaging sequences. The volumetric assessment of CC was conducted on coronal oblique T1-weighted images. The imaging parameters of the T1-weighted Magnetization Prepared Rapid Acquisition Gradient Echo (MPRAGE) sequence were as follows: repetition time (TR): 2,400 ms, echo time (TE): 3.54 ms, the field of view (FOV): 240 mm, matrix = 200, flip angle = 7, slice thickness: 1.2 mm, voxel size: $1.3 \times 1.3 \times 1.2$ mm.

### Segmentation and volumetric analysis using MRICloud

CC and its parts volume were measured using the MRICloud (https://braingps.mricloud.org/). MRICloud use multi-atlas label-fusion (MALF) algorithms for normal normalization and registration. There are two major components in the algorithms: first is image registration and second is the multi-atlas label fusion process. In the registration step, atlases are transformed to the target image using large deformation diffeomorphic metric mapping (LDDMM). All MRIs are corrected and linearly aligned to the JHU-MNI atlas space according to MRICloud. First, the atlases are linearly aligned using LDDMM (Ceritoglu et al., 2009) warped to the subject image and then the multi-atlas fusion algorithm (Yeshu et al., 2022) is applied. The whole brain was segmented into 289 structures including the corpus callosum, cerebellum, fourth ventricle, brain stem, etc., (Djamanakova et al., 2014; Soysal et al., 2022) using web-based software MRICloud (https://mricloud.org). The two atlases (adult_286 labels_11atlases_V5L and JHU multi-atlas inventories with 286 defined structures) were used for processing the MRI data (Otsuka et al., 2019; Yılmaz et al., 2020). The individual volumes of CC parts (genu, body, and splenium) were automatically parcellated and calculated using MRICloud. Although the corpus callosum consists of four regions (rostrum, genu, truncus, and splenium), CC is divided into three regions using MRICloud parcellation. The rostrum is included in the genu region according to MRICloud.

High-resolution 3D T1-weighted MRI data of all subjects in DICOM format was uploaded from PACS to a 64-bit computer (Windows 10). Image (IMG) and header (HDR) files were created from these DICOM format images using Mricron software (Czeibert et al., 2019). These files were uploaded to the "T1-Multiatlas" area in the
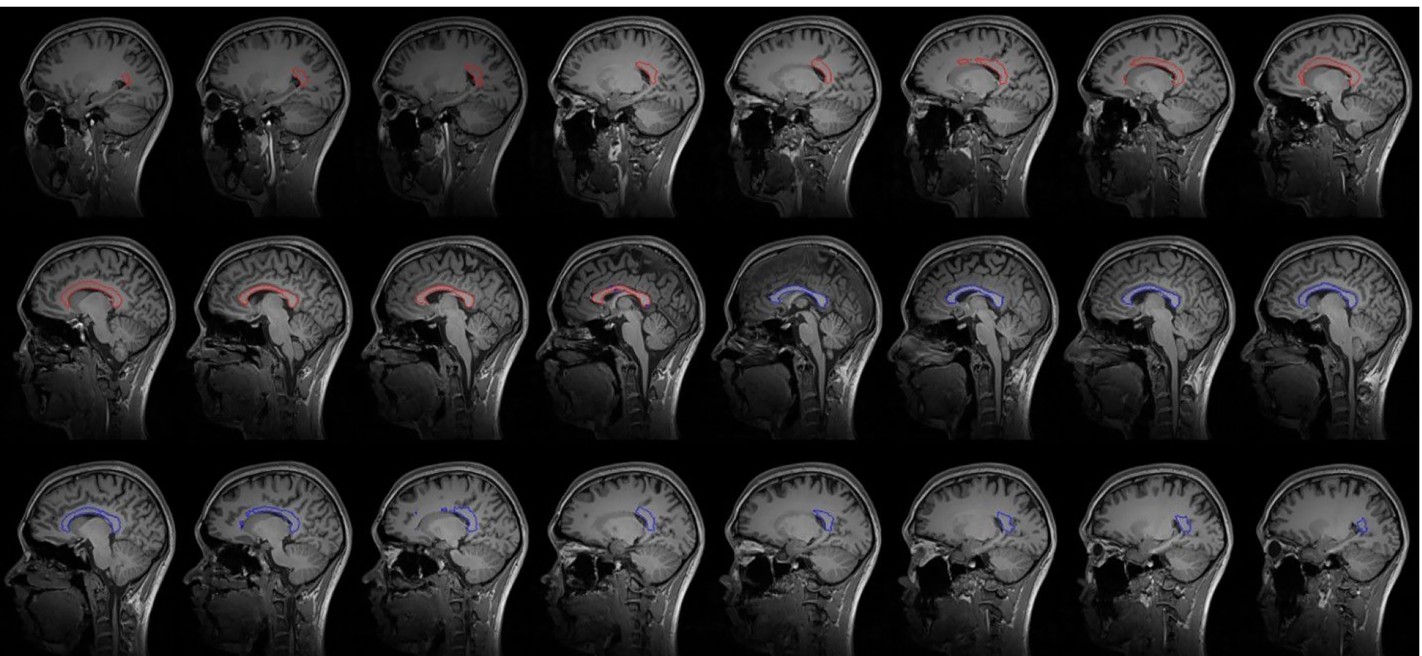

**Figure 1 Segmentation with MRICloud in the BrainGPS of of CC and subregions.**

"Segmentation" section of the MRICloud in the BrainGPS. The best atlases were selected and submitted. The whole-brain parenchyma was automatically divided into 2873289 anatomical regions (*Işıklar et al., 2023*). The segmentation maps obtained in the "View Result" section in "My Job Status" were visually inspected for quality control of the corpus callosum and its subregion (Fig. 1).

The volumes of the mesencephalon, telencephalon, metencephalon, diencephalon, myelencephalon, and CSF were summed to calculate total brain volume. We got six volume measurements as genu, body, and splenium of CC (right and left). The CC subregion and total volume measurements were normalized by dividing them by total brain volume per patient. These adjusted values were used in ML analysis (Fig. 2).

## Age and sex classification using machine learning

The classification of age and results was performed using the Classification Learner Tool in MATLAB 2020b (MathWorks, Natick, MA, USA). Six volume measurements were used as predictors: genu, body, and splenium of CC (right and left). Two ML models were created: one with a categorical outcome (*i.e.*, young (age <= 25) and adult (age > 25) and another with a categorical outcome (*i.e.*, sex: male and female).

The classification performances were tested for both outcomes (25 ML algorithms) in the domains of decision tree, linear and quadratic discriminant, logistic regression (for binary outcomes only), naïve Bayes, support vector machine (SVM), nearest neighbor classifiers (KNN), and Ensemble classifiers. Each of these classifiers is included in the MATLAB Classification Learner Toolbox. They provide supervised ML, utilizing the

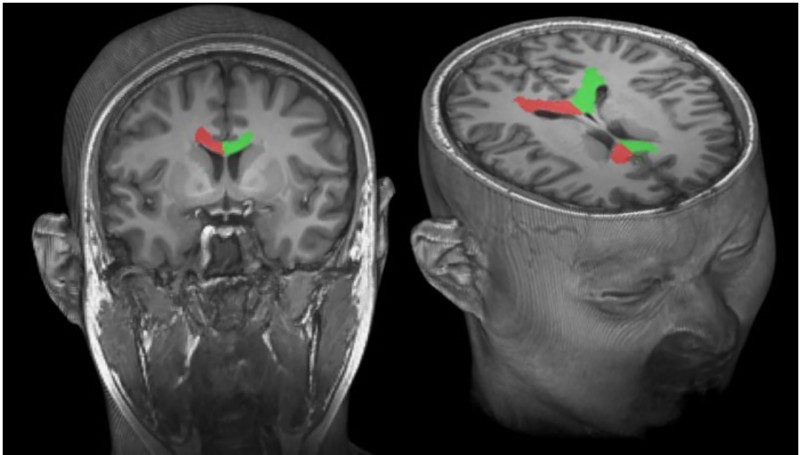

**Figure 2** **T1-weighted MRI.** 3D visualization of CC and subregions.

labeled data. Each classifier evaluates the data using distinctive computing methods with varying computational speeds. All the available classifiers were used for evaluation.

## Model evaluation and statistical analysis

The data were prepared as two tables: volume measurements and age; volume measurements and sex. Ten-fold cross-validation was done to evaluate the performance of ML models. In other words, each data table was partitioned into ten datasets. For each dataset (validation dataset), the model was trained using the remaining datasets (training dataset). Then, model performance was assessed using each validation dataset. Lastly, validation error was calculated by averaging validation error over all datasets. Classification accuracy for age and sex and receiver operating characteristics (ROC) were obtained.

This repeated cross-calibration technique yields a good estimate of the accuracy of the model and is recommended for small data sets.

## RESULTS

In this study, this is the gender and age groups and their distribution in this study. Young ($n$ = 52) ((male ($n$ = 17), female ($n$ = 35)), adults ($n$ = 250) (male ($n$ = 102), female ($n$ = 148)).

The volumes of CC parts were obtained based on the T1 segmentation (Fig. 2). Figure 1 shows a visualization of the CC segmented by MRICloud, overlaid to a sagittal slice of a T1-weighted MRI. 3D visualization of CC is shown in Fig. 3. We also reported the sensitivity and specifcity in the results section.

Gender classification (male *vs*. female) achieved a sensitivity of 88% and a specificity of 28%. For age classification (young *vs*. adult), sensitivity was 99% and specificity was 13%.

List of machine learning algorithms used in classification and their accuracies. The fine Gaussian SVM model showed the best performance in classifying age and gender (Table 1).

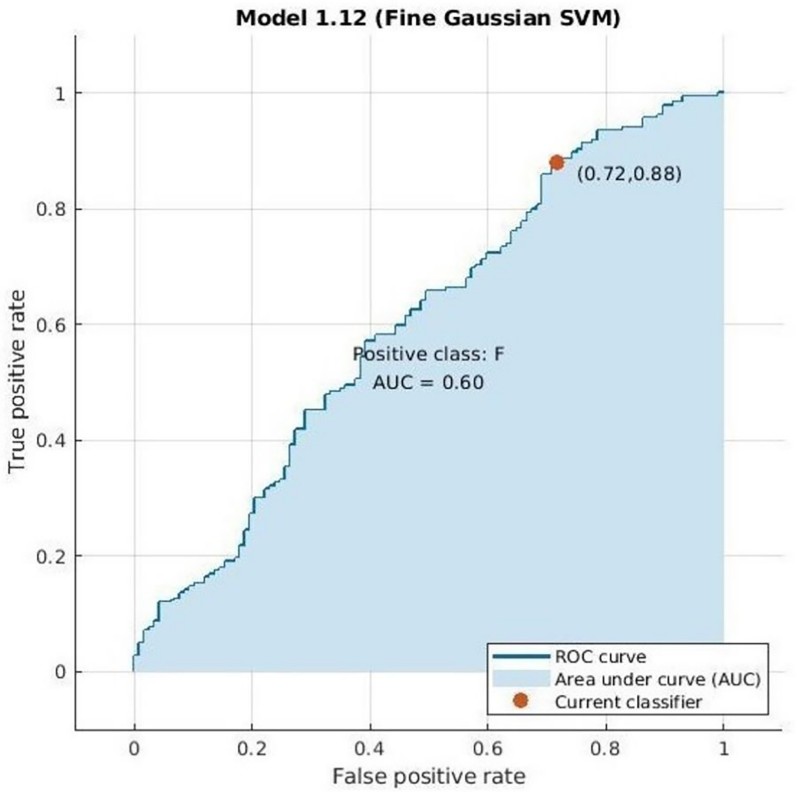

**Figure 3 The AUC of the ROC curves in predicting age as two categories.**

**Table 1 The list of machine learning algorithms used in classification and their accuracies.**

| Number | Algorithm | Gender (Male/Female) classification accuracy (%) | Age (Young/Adult) classification accuracy (%) |
|---|---|---|---|
| 1 | Fine Tree | 49.5 | 75.7 |
| 2 | Medium Tree | 52.8 | 76.1 |
| 3 | Coarse Tree | 60.1 | 78.7 |
| 4 | Linear Discriminant | 61.5 | 81.1 |
| 5 | Quadratic Discriminant | 57.1 | 78.7 |
| 6 | Logistic Regression | 60.5 | 80.4 |
| 7 | Gaussian Naive Bayes | 65.4 | 77.7 |
| 8 | Kernel Naive Bayes | 63.1 | 78.1 |
| 9 | Linear SVM | 60.1 | 82.4 |
| 10 | Quadratic SVM | 61.1 | 80.7 |
| 11 | Cubic SVM | 57.8 | 75.7 |
| 12 | Fine Gaussian SVM | 65.4 | 83.7 |
| 13 | Medium Gaussian SVM | 60.5 | 82.4 |
| 14 | Coarse Gaussian SVM | 60.5 | 82.4 |
| 15 | Fine KNN | 57.8 | 74.4 |
| 16 | Medium KNN | 61.1 | 82.7 |

| Table 1 (continued) | | | |
|---|---|---|---|
| Number | Algorithm | Gender (Male/Female) classification accuracy (%) | Age (Young/Adult) classification accuracy (%) |
| 17 | Coarse KNN | 63.5 | 82.4 |
| 18 | Cosine KNN | 61.5 | 82.1 |
| 19 | Cubic KNN | 60.8 | 81.7 |
| 20 | Weighted KNN | 61.5 | 80.7 |
| 21 | Boosted | 58.1 | 78.1 |
| 22 | Bagged | 61.5 | 78.1 |
| 23 | Subspace Discriminant | 61.8 | 82.7 |
| 24 | Subspace KNN | 56.1 | 78.1 |
| 25 | RUSBoosted Trees | 53.8 | 63.5 |

**Note:**
The Fine Gaussian SVM model yielded the best performance in classifying age and gender. (SVM, Support Vector Machien; KNN, K-Nearest Neighbor).

## Sex classification

Twenty-five ML algorithms were tested based on the right and left volumes of the genu, body, and splenium of CC to predict sex (male (117); female (184)). The best classification results were obtained using fine Gaussian SVM with an accuracy of 65.4% and an area under the curve (AUC) of the ROC curve of 0.60. Figure 4 demonstrates the AUC of the ROC curves for the SVM model.

## AGE

When the ML algorithms were tested to categorize the age as young and adult, the best classification results were obtained using the fine Gaussian SVM with an accuracy of 83.7% and an AUC of the ROC curves of 0.67 for young and rest. Figure 5 demonstrates the AUC of the ROC curves in predicting age as two categories.

## DISCUSSION

The CC has nearly 200 million fibers and is the largest white matter that connects each hemisphere in the human brain (*Huang et al., 2005*; *Tomasch, 1954*). Its function is to integrate information and mediate behaviors (*Hinkley et al., 2012*). This study hypothesizes that ML algorithms can classify age and sex based on volumetric data of the CC obtained from MRI images. The CC is a critical structure that facilitates communication between the two hemispheres of the brain, and its volume may show significant associations with demographic factors such as age and sex. The morphological differences in relation to age and sex have been a popular topic of investigation for decades, as there is tremendous potential in relating the structural differences to pathological, behavioral, cognitive, and functional age- and sex-related differences. It is known that brain changes with aging. In addition, there are sex-related differences in human brain too. So far, we have been studying these using images and making volumetric comparisons.

We never tried if ML algorithms can detect these differences themselves. We started simple. We calculated the volumes of CC and trained ML algorithms using them as inputs.

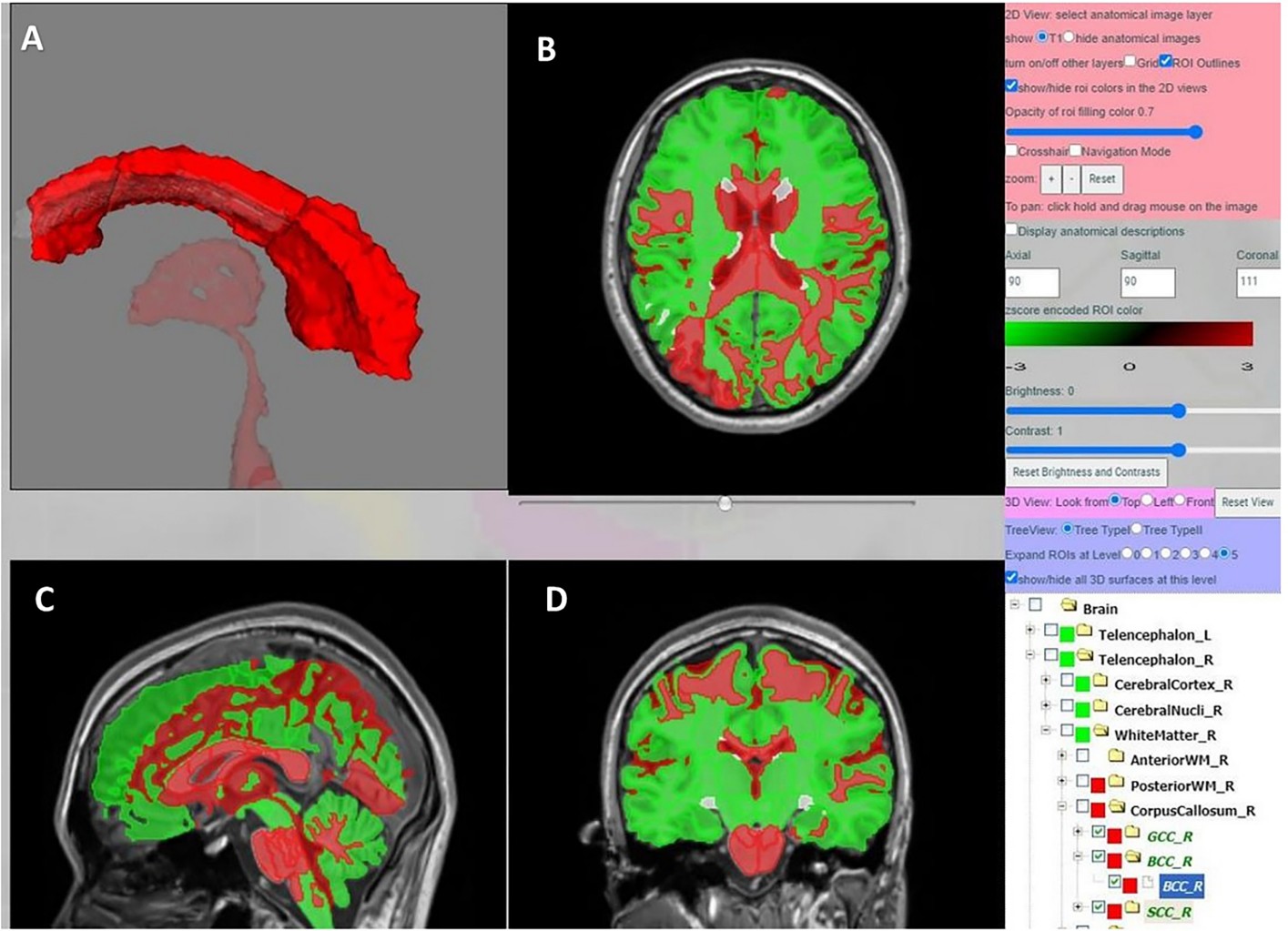

**Figure 4** **Three-dimensional representation of the CC and its subregions (genu, trunk and splenium) after parcellation using MRICloud.** The CC consisting of four regions (rostrum, genu, trunk and splenium) was divided into three regions using MRICloud. The rostrum is included in the genu region according to MRICloud. (A) The anatomical structures of the corpus callosum are shown in three-dimension. (B) Axial. (C) Sagittal. (D) Coronal.

We unfortunately could not get a huge success but we saw a possibility. Our study shows that ML has potential to look at human brain in a way that is not possible before.

Many studies have shown that age, gender, prenatal and perinatal trauma, hypoxic injuries, variation, genetic factors, environmental factors, ethnicity, demyelinating disorders, and neurological and some psychiatric diseases such as Alzheimer's disease, Williams syndrome, Down's syndrome and bipolar disorder are associated with the changes in the shape or size of CC (*Acer & Gürlek, 2023*).

Despite advances in neuroimaging and ML, there are still significant gaps in our understanding of how demographic factors like age and sex influence brain structures. This study aims to fill these gaps by providing quantitative evidence. Developing non-invasive, accurate diagnostic tools based on MRI data and ML can revolutionize the field of

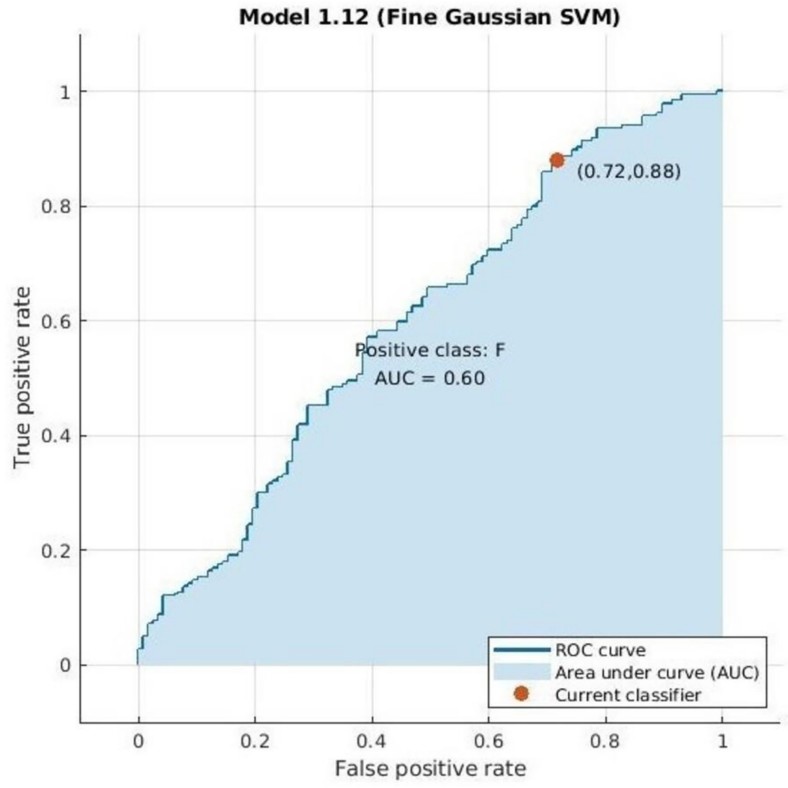

**Figure 5 The AUC of the ROC curves for the SVM model.**

neurology. This study could lead to practical applications in clinical settings, improving diagnostic accuracy. The findings are expected to contribute to the early diagnosis of neurological diseases and to neuroscience research.

We tested ML algorithms to classify age and sex based on volumetric CC data. To our best knowledge, this has not been explored before in literature. For segmentation and volume calculation of CC, we used MriCloud which is an online software (https://mricloud.org/) (*Ma et al., 2015*; *Djamanakova et al., 2014*). Comparing it to single-atlas techniques, its segmentation accuracy with multi-atlas fusion was proven (*Artaechevarria, Munoz-Barrutia & Ortiz-de-Solorzano, 2009*). MriCloud allowed us to examine the subregions of CC by providing us with the volumes of genu, body, and splenium. The previous reports in the literature lacked these details by focusing only on a single median slice or calculating volume with imprecise or impractical methods.

The a few of CC volumetric studies have focused on differences among special groups, gender differences, and age differences. *Acer et al. (2018)* found that most of the CC volume was smaller in musicians than in a control group, except for the left genu and right splenium, but the difference was not statistically significant (*Acer et al., 2018*). More recently, *Soysal et al. (2022)* stated that all volumes of the CC were significantly greater in males than females, and the genu and splenium volumes were significantly greater in the left hemisphere in both sexes (*Soysal et al., 2022*). Most recently, *Işıklar et al. (2023)* studied

age- and sex-related changes in volumetric development of CC from birth to 18 years old (696 subjects). They used 3D-T1-weighted sequence for their retrospective study and they calculated the genu, body, splenium, and total volume of CC using MRICloud. They found that the total CC volume has six developmental periods 0 years, 1, 234, 539, 10,316, and 17,318 years; also, they found that all volume measurements of total CC were highly positively correlated with age according to the correlation coefficient analysis (*Işıklar et al., 2023*). *Giuliano et al. (2018)* compared the CC volume in ASD and controls using FreeSurfer v5.1, a widely-used semi-automated brain-segmentation methodology. They found that the CC total volume positively correlates with age in control groups. *Sugijono et al. (2020)* calculated corpus callosum index (CCI) and used FreeSurfer software, 30 MS 30 patients. They found that The CCI was positively correlated with age of onset (*Giuliano et al., 2018*).

Studies have focused on the relationship between area measurements of midsagittal CC and cortical lateralization or asymmetries of various brain structures. *Luders et al. (2006)* performed thickness and area measurements for CC asymmetry analysis on 1 mm thick parasagittal sections 6 mm away from the mid-sagittal CC (*Luders et al., 2006*). According to the asymmetry coefficient of the area measurements of the CC, they found right lateralization in the region up to the isthmus and left lateralization in the isthmus and splenium (*Luders et al., 2006*, *2003*; *Herron, Kang & Woods, 2012*) found that the callosum showed increased in length and reduced in thickness and area with age an analysis of age-related changes. *Prendergast et al. (2018)*, automatically identified the midsagittal plane and obtained CC subregion measures based on *Witelson (1989)* and *Prendergast et al. (2018)*. Measurements of the CC's area, perimeter, length, circularity, and subregional area were made by *Stapledon (1998)*. They discovered that descriptive linear correlations and peak values for CC subregions between ages 32 and 45 were congruent with areas that had undergone more significant developmental changes. Also, they observed varying age related changes across the lifetime between males and females in the CC subregion corresponding to the genu, as well as CC circularity. *Shrestha et al. (2022)* stated that the whole corpus callosum and rostrum was significantly correlated with white matter changes. Similarly, in bivariate regression analysis, white matter changes were strongly correlated with rostrum.

*Genc et al. (2018)* investigated microstructural metrics of the CC and assessed the effect of age, and sex. They used DTI and calculated apparent fiber density metrics. They discovered that apparent fiber density showed a substantial correlation with age across the corpus callosum, especially in its posterior region. Additionally, compared to males, females had much higher apparent fiber density, especially in the anterior splenium and posterior genu. Additionally, they discovered that the splenium was the only organ with age-matched pubertal group variations (*Genc et al., 2018*). Corpus callosum is also a key structure in schizophrenia, Alzheimer's disease (AD), bipolar disorder, epilepsy, autism, psychosis, and unipolar depression (*Sullivan, Rosenbloom & Desmond, 2001*; *Guenette et al., 2018*; *Arnone et al., 2008*; *Arnone et al., 2008*). A quantitative survey using the ABIDE dataset reported an increase in brain volume and a reduction in CC area in autism

spectrum disorder (*Zaidel & Iacoboni, 2003*). Some studies have reported a decrease in the size of the CC in schizophrenia (*Arnone et al., 2008*; *Wood et al., 2006*). Studies on psychosis and CC volume reported that a lower volume of CC or lower CC-to-brain ratio was present in psychosis groups (*Rossell et al., 2003*). *Agdanlı et al. (2022)* found that the CC volume was lower in patients with first-episode psychosis than in healthy controls. Some studies also found atrophy of the corpus callosum in AD (*Prabakar & Porkumaran, 2012*). The size of the spleen and isthmus was found to have decreased by *Waite et al. (2005)*. ASD was also associated with a decrease in the splenium, genu, and rostrum of the corpus callosum, according to *Chung et al. (2004)* and *Sharif & Khan (2022)* investigated a system for the automatic identification of ASD utilizing data from T1-weighted MRI scans of the ABIDE dataset and features extracted from the CC and intracranial brain volume. They discovered that the achieved recognition rate fell between 55% and 65% (*Chung et al., 2004*; *Sharif & Khan, 2022*). Although the aforementioned studies examined CC volume and its association with cognitive and clinical parameters, ML has not been used to explore its association with these parameters. ML offers exciting opportunities in neuroimaging to study the brain in a way that has not been done before and/or validate our existing knowledge. We explored if we can show the volumetric change of CC and its subregions due to age and sex using ML techniques. We used 25 ML algorithms and trained them using the right and left volume measurements of the genu, body, and splenium of CC to predict sex: female and male. The best classification results were obtained using fine Gaussian SVM with an accuracy of 65.4%. When the age was reduced to two categories: young and adult, the best classification results were obtained using the fine Gaussian SVM with an accuracy of 83.7%. Our findings show some association between the volumes of CC and sex and age. Fine Gaussian SVM appears to be a working model for age and gender classification. SVM separates classes by finding the hyperplane that maximizes the margin between classes. The boundaries in the SVM feature space seem to correlate with age and gender.

There are some limitations in our study. First, our results are based on a small sample size. ML requires big datasets to train. Our cohort was 300 subjects in this pilot study. Second, the number of females and males was not equal which might bias the ML algorithms. Third, the algorithms used were internally validated using repeated cross-validation protocols that divide the data into random training and testing sets. We did not test the algorithms on external datasets. However, incorporating additional data might strengthen the predictive ability of the algorithms.

## CONCLUSION

This study demonstrates the potential of machine learning to identify age and sex effects on corpus callosum volume, achieving classification accuracies between 65% and 83%. While larger datasets are needed to enhance performance, our findings highlight the value of ML in examining CC volume changes, particularly relevant to neurological disorders like schizophrenia and psychosis. These insights contribute to the development of more accurate demographic classification models using brain structural data.

## ABBREVIATION LIST

| | |
|---|---|
| **CC** | Corpus callosum |
| **MRI** | Brain magnetic resonance imaging |
| **KNN** | k-Nearest Neighbor |
| **SVM** | Support Vector Machine |
| **AD** | Alzheimer's disease |
| **ML** | Machine learning |
| **AI** | Artificial intelligence |
| **MMSE** | A Mini-Mental State Examination |
| **CSF** | Cerebrospinal Fluid |
| **ROC** | Receiver operating characteristics |

### Funding

The authors received no funding for this work.

### Competing Interests

The authors declare that they have no competing interests.

### Author Contributions

- Handan Soysal conceived and designed the experiments, performed the experiments, analyzed the data, prepared figures and/or tables, authored or reviewed drafts of the article, concept design supervision resources materials data collection and/or processing critical review, and approved the final draft.
- Niyazi Acer conceived and designed the experiments, performed the experiments, analyzed the data, performed the computation work, prepared figures and/or tables, authored or reviewed drafts of the article, concept design supervision resources materials data collection and/or processing critical review, and approved the final draft.
- Meltem Özdemir conceived and designed the experiments, prepared figures and/or tables, analysis and/or interpretation literature search writing manuscript critical review, and approved the final draft.
- Kazim Gumus conceived and designed the experiments, performed the experiments, analyzed the data, performed the computation work, prepared figures and/or tables, authored or reviewed drafts of the article, analysis and/or interpretation literature search writing manuscript critical review, and approved the final draft.

### Data Availability

Code and raw data are available in the Supplemental Files.

## Supplemental Information

Supplemental information for this article can be found online at http://dx.doi.org/10.7717/peerj-cs.3179#supplemental-information.

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
