# Peer review of "Machine learning for detecting age and sex effects on corpus callosum volume"

_PeerJ Computer Science, doi:10.7717/peerj-cs.3179_

## Round 0.1 · original submission · Major Revisions

Please target in detail all the criticisms and modify the manuscript accordingly.

**Language Note:** The review process has identified that the English language must be improved. PeerJ can provide language editing services - please contact us at [email protected] for pricing (be sure to provide your manuscript number and title). Alternatively, you should make your own arrangements to improve the language quality and provide details in your response letter. – PeerJ Staff

Reviewer 1 ·

Basic reporting

-

Experimental design

-

Validity of the findings

-

Additional comments

The study is presented in a clear, concise, and effective way. Although the topic addressed is of great interest, the results obtained in terms of overall performance (AUC) are rather poor. Therefore, the study will have to be repeated on larger datasets, and this must be highlighted among the limitations of the study.

Moreover, in order to be published, some parts need to be significantly improved. Below are my comments:

1. I ask the authors to improve the caption description of Figure 2, because for me it is not so clear.

2. I suggest the authors report in the text the results achieved using all the classifiers; otherwise, it seems useless to include them in the work.

3. The folds used in all cross-validation analyses should be balanced by gender and age, as the numbers are unbalanced (52 young vs 250 adult, and 102 M vs 148 F). Due to this imbalance, the reliability of the results needs to be further proved. Furthermore, I would suggest that the authors repeat the cross-validations several times to attribute uncertainty to the AUC and verify that it is actually different from the chance level (0.5).

4. I’m sorry, but in my version, Table 1 is missing.

5. Lines 169-170: I wouldn't talk about the success of the models because the AUC is quite low.

Cite this review as

·

Basic reporting

Generally understandable. But typos are present, some sentences are duplicated, and others need to be rephrased. I recommend proofreading or professional editing.

The method section reported 5-fold validation, but other sections reported 10-fold validation.
Some axes lack units.

I recommend adding neuroimaging references on sex/age prediction from structural MRI.
Age and Sex classification using Artificial Intelligence is used, but I recommend using "Machine learning analysis", as it's more sound.

Experimental design

Design is clear and within the aims and scope of the journal.

Other measures, such as balanced accuracy, sensitivity, and specificity, should be reported.

Rationale for defining the young vs elderly group was not referenced nor elaborated.

Validity of the findings

The author claims that the "ML can be applied"; however, since the accuracy lies around 0.6, it's considered modest.

Cite this review as

---

## Round 0.2 · Minor Revisions

Please address some remaining comments properly.

·

Basic reporting

Manuscript language has been overall improved. I still recommend to explain and add a reference to how the groups were defined as young and adult, meaning how did the cut off value of 25 year old was defined. Also, the participant section shall be in the results, not in the methods. In the abstract, in the method section, remove the participant and age/gender information and insert it in the result section of the abstract instead. I also recommend reporting all of the ML models utilized, not only SVM. I still recommend to report the sensitivity, specificity and balanced accuracy within the result section, not only on the figure. The result section is too short as well.

Experimental design

Well designed.

Validity of the findings

Further explain the 25 ML learning algorithms used, even if briefly explained.

Additional comments

Overlapping between result and method sections as previously mentioned.

Cite this review as

---

## Round 0.3 · accepted · Accept

The authors seem to have addressed my previous comments properly.